# A Single-Celled Metasurface for Multipolarization Generation and Wavefront Manipulation

**DOI:** 10.3390/nano12234336

**Published:** 2022-12-06

**Authors:** Ruonan Ji, Xin Guo, Zhichao Liu, Xianfeng Wu, Chuan Jin, Feng Liu, Xinru Zheng, Yang Sun, Shaowei Wang

**Affiliations:** 1School of Physical Science and Technology, Northwestern Polytechnical University, Xi’an 710129, China; 2Science and Technology on Low-Light-Level Night Version Laboratory, Xi’an 710065, China; 3State Key Laboratory of Transient Optics and Photonics, Xi’an Institute of Optics and Precision Mechanics of CAS, Xi’an 710119, China; 4State Key Laboratory for Infrared Physics, Shanghai Institute of Technical Physics, Chinese Academy of Sciences, Shanghai 200083, China

**Keywords:** metasurface, polarization generation, wavefront shaping, Aharonov–Anandan (AA) geometric phase

## Abstract

Due to their unprecedented ability to flexibly manipulate the parameters of light, metasurfaces offer a new approach to integrating multiple functions in a single optical element. In this paper, based on a single-celled metasurface composed of chiral umbrella-shaped metal–insulator–metal (MIM) unit cells, a strategy for simultaneous multiple polarization generation and wavefront shaping is proposed. The unit cells can function as broadband and high-performance polarization-preserving mirrors. In addition, by introducing a chiral-assisted Aharonov–Anandan (AA) geometric phase, the phase profile and phase retardation of two spin-flipped orthogonal circular polarized components can be realized simultaneously and independently with a single-celled metasurface via two irrelevant parameters. Benefiting from this flexible phase manipulation ability, a vectorial hologram generator and metalens array with spatially varying polarizations were demonstrated. This work provides an effective approach to avoid the pixel and efficiency losses caused by the intrinsic symmetry of the PB geometric phase, and it may play an important role in the miniaturization and integration of multipolarization-involved displays, real-time imaging, and spectroscopy systems.

## 1. Introduction

Polarization is an inherent property of light which is essential for many applications, such as polarization detection and imaging, nonlinear optics, three-dimensional displays, and data storage and transfer. In many polarization-involved applications, such as polarization detection or imaging for dynamic or living targets, multipolarization displays, and polarization multiplexing, multipolarization generation and manipulation are required. Linear polarizers and waveplates are traditional optical elements for polarization generation and manipulation. However, with a single polarizer, only one polarization can be filtered, and the generation of circular polarization requires cascading linear polarizers and quarter waveplates. Therefore, in general, lots of optical elements for polarization generation and wavefront manipulation are needed for a multipolarization-involved system, which makes them bulky and complicated. How to obtain miniaturized optical elements with functions of multipolarization generation and manipulation is significant for the development of multipolarization-involved systems.

The emergence of metasurfaces has provided a promising and artful approach to solving the above problem. Benefiting from an unprecedented and flexible capability in the manipulation of the intensity, polarization, and phase of light [1,2,3], recent years have witnessed the development of various metasurface-based devices, such as metalenses [4,5,6,7], vortex generators [8,9,10,11,12,13], holograms [2,9,14,15,16,17], metagratings [1,18], nanoprints [19,20,21,22], and so on. Among them, in terms of polarization conversion, imposing different phase retardations on orthogonal linear/circular polarizations is the general design strategy [10,23,24]. The propagation phase and geometric phase are two basic phase types used in the design of a metasurface [25]. Therein, the geometric phase is introduced during the SU(2) manipulation with the polarization state on the Poincare sphere, namely, in the process of the circular polarization conversion (spin flipping). Moreover, different from the propagation phase, the geometric phase is independent of the frequency and only related to the geometric path of evolution, i.e., the angle of rotation of the polarization ellipse from the perspective of Coriolis shift. Thus, the geometric phase is favored by researchers due to its advantage of nondispersive modulation on phase, and the Pancharatnam–Berry (PB) geometric phase is the first proposed and most widely used geometric phase [23,26,27]. When contrarotating the anisotropic structure by angle *θ*, the angle of rotation of the polarization ellipses corresponding to the right-handed circular polarized (RCP) and left-handed circular polarized (LCP) phases are the same values but in opposite directions. Then a PB phase shift of ±2*θ* will be introduced to the spin-flipped component, and herein, the RCP light takes a negative sign while the LCP light takes a positive sign. Such intrinsic symmetry makes it impossible to achieve arbitrary control of the polarization state with a single-celled scheme. Once the phase distribution of LCP light is set, the one for RCP light is fixed. For example, if the metasurface phase profile under LCP incidence is set for focusing, the metasurface will diverge the RCP incidence. As a compromised solution, arranging two or more nanostructures respectively manipulating the LCP and RCP localized phases as a supercell to form an interleaved metasurface is widely employed [23,24]. However, such a configuration may bring an unavoidable loss of pixels and low energy efficiency.

Very recently, another geometric phase, the nonadiabatic Aharonov–Anandan (AA) phase was introduced in the design of a metasurface [9,10,28]. In the scheme of the nonadiabatic AA phase, the structure parameters are changed to rotate the polarization ellipse and therefore introduce geometric phase shift. In this way, it provides a promising approach to decouple the angle of rotation of the polarization ellipses respectively corresponding to the RCP and LCP phases. Therefore, as a kind of geometric phase, the AA phase has attracted much attention as it not only possesses the advantage of nondispersive phase control but also has the potential to artfully avoid the intrinsic symmetry of LCP and RCP phases [9,28,29]. Benefiting from such merit and the spin-dependent response of the chiral structure, an umbrella-shaped metasurface was proposed as an example to verify the validity of the spin-decoupled AA phase manipulation mechanism. Then, a combined geometric phase metasurface for dual manipulations was proposed to realize simultaneous holography/vortex generation and polarization rotation [10]. While in previous work, the AA phase was introduced to break the intrinsic symmetry of the PB geometric phase. As the change of the PB phase led to a chain response of both phases of LCP and RCP, the desired AA phase profile needed to be obtained by some calculation. In this work, analogous to the optical activity process in traditional media, a spin-decoupled AA phase manipulation mechanism was further applied to achieve broadband and achromatic optical activity, and a new design scheme of single-celled metasurfaces which perform bifunctional polarization control and wavefront shaping was proposed. The design of the metasurface was further simplified, as only the AA geometric phase was applied, and the phase profile of the LCP and RCP components were independently controlled by two irrelevant parameters. Based on the proposed strategy, a vectorial hologram generator and metalens for multiple polarization generation and focusing were proposed, which may have a far-reaching influence on the miniaturization of multipolarization-involved optical systems.

## 2. Working Principle and Unit Cell Design

According to the theory of optical activity proposed by Fresnel, when a linear polarized (LP) light incident is in a media with optical activity, the LP state can be decomposed into two orthogonal circular polarization states with equal amplitude. The polarization rotation angle of the transmitted light is determined by the phase retardation *δ* between the LCP and RCP components. Such a relationship can be expressed as Equation (1) with a Jones vector [10]:(1)$$|LP\rangle =\left[\begin{array}{c} cos\varphi \\ sin\varphi \end{array}\right]=\frac{\sqrt{2}}{2}e^{-i\varphi }\left(e^{i2\varphi }|LCP\rangle +|RCP\rangle \right)=\frac{\sqrt{2}}{2}e^{-i\varphi }\left(e^{i\delta }|LCP\rangle +|RCP\rangle \right)$$ where *φ = δ/2* is the polarization rotation angle, and |LP〉, |LCP〉, and |RCP〉 are the polarization states of LP, LCP, and RCP light, respectively.

Analogous to the optical activity process in traditional media, in this paper, a metasurface with meta-atoms capable of manipulating the phases of RCP and LCP components independently was introduced. Since the geometric phase is generated during the spin-flipping of circularly polarized light, the meta-atom was designed as a three-layered MIM structure (see Figure 1a) with broadband and high circular polarization preserving ability (in the reflection scheme, spin-flipping components correspond to circular polarization preserving components, i.e., R_LL_ and R_RR_; here, the first and second subscripts represent the handedness of the reflected and incident light, respectively) [9]. The optical properties of the meta-atoms were simulated based on the finite element method (FEM). Detailed information about the simulation method is described in the Appendix B, and a discussion on the efficiency of the designed meta-atom (see Appendix A) can be found in Appendix A. The umbrella-shaped nanostructure on the top introduced a spin-dependent response due to the chiral configuration, and our previous study showed LCP and RCP incident light mainly interact with the left and right parts of the nanostructure, respectively. Moreover, different from normal anisotropic structures like nanobrick, the principal axis of the umbrella-shaped structure was determined by the lengths of the left and right arms. These features provided an effective approach to generate spin-independent rotation of the polarization ellipse without rotating the structure. As shown in Figure 1b,c, when the value of *β* is fixed, the increase of *α* will bring an evolution of *φ*_LL_ (the phase of R_LL_) varying from 0° to 360° while *φ*_RR_ (the phase of R_RR_) remains nearly constant at the same wavelength. A similar phenomenon can be observed when *α* is fixed and *β* changes, except that *φ*_LL_ and *φ*_RR_ are interchanged. Moreover, for structures with the same *α* and *β*, the phase retardation *δ* is stable in the whole operation band. In the other words, when an x-LP (i.e., LP polarization with polarization direction along the *x*-axis) incident interacts with the meta-atom, an achromatic phase retardation *δ* will introduce the two spin-flipped reflection components. Meanwhile, the intensity difference between the two components (R_LL_ and R_RR_) is small enough to be ignored. Thus, these two components are coupled in the near-field and contribute to broadband and achromatic optical activity effects. By synchronously controlling *φ*_LL_, *φ*_RR,_ and *δ*, it is possible to independently control the wavefront as well as the polarization state of reflection in a single-celled metasurface.

## 3. Metasurface for Vectorial Holography

Vectorial metaholography refers to a hologram diffracts holographic images with spatially varying polarization states, as shown in Figure 2a. Compared to conventional scalar holography, vectorial holography records intensity, phase, and polarization features, which largely increases the information capacity carried by the light wavefront [30]. In previous reports, due to the intrinsic spin symmetry of the PB geometric phase, a metamolecule composed of at least two orthogonal meta-atoms was needed to control the phase retardation, leading to the unavoidable loss of pixels [31]. Benefiting from the flexible phase control of the proposed single-celled metasurface, a vectorial metahologram containing 0°, 45°, 90°, and RCP polarization states was presented experimentally.

The design process of the metasurface is shown in Figure 2b. Based on the Gerchberg–Saxton (GS) algorithm for the phase-only hologram, the phase profiles *Φ*_g1_, *Φ*_g2_, *Φ*_g3_, and *Φ*_g4_ of the four target images were calculated. Phase gradient *Φ*_d_ along the *x*-axis providing a 20° deflection was introduced to avoid the influence of the spin-preserving components. Then, the four-phase distributions were respectively encoded onto the single-celled metasurfaces M_1_, M_2_, M_3_, and M_4_, and interleaved to form the final sample. Herein, in the design of M_1_, M_2_, and M_3_, the value of *φ*_RR_ at each pixel was modulated by changing the central angle *β* of each unit cell, and *φ*_RR_ = *Φ*_gi_ + *Φ*_d_ (*i* = 1,2,3). In addition, the value of *φ*_LL_ was modulated by changing the central angle α, and *φ*_LL_ = *Φ*_gi_ + *Φ*_d_ + 2*δ* (i = 1, 2, 3), where *δ* is the polarization rotation angle regarding the polarization direction of the incident light; the values are 90, 0, and 45, respectively. As for M_4_, central angle α was fixed at 60° while *φ*_RR_ = *Φ*_g4_ + *Φ*_d_ was encoded in each central angle *β* of the metasurface.

The designed metasurface was fabricated via a typical lift-off process. Figure 2c,d shows the SEM photo of the fabricated sample with 400 × 400 meta-atoms and the experimental setup of the vectorial holography testing. A fiber laser operating at 1.55 µm was used as the light source; after being collimated, the light passed through a polarizer to generate a normally incident LP light. The target images were reconstructed via the Fourier transform of the objective lens, and the polarization states of the images were examined by polarizer or the combination of a polarizer and quarter waveplate. More detailed fabrication and testing information can be found in the Appendix B. Due to the limited computer memory, the four metasurfaces (M_1_–M_4_) for different target reconstructions were simulated separately, and the dimension of the arrays was chosen as 50 × 50. As shown in Figure 3a, the simulated results verify the broadband working ability, as similar reconstructed images of the targets can be observed from 1.2 µm to 2 µm. In the process of diffraction, the size of the reconstructed image is proportional to the wavelength; thus, it can be found that the reconstructed image size gradually increases from 1.2 to 2 µm. Then, the feature of spatially varying polarizations was verified by an experimental polarization-sorting test. Upon rotation of polarizer P_2_, the reconstructed image of circularly polarized targets T_4_ was almost unchanged while the linearly polarized targets (T_1_, T_2_, and T_3_) showed obvious polarization characteristics. Specifically, as shown in Figure 3b, when the angle between polarizer P_1_ and P_2_ changed from 45° to −45°, the reconstructed image of T_3_ changed from clear to nearly disappeared. After adding a quarter waveplate before P_2_ and rotating P_2_, it can be observed that the reconstructed image of T_1_-T_3_ was almost unchanged while the reconstructed image of T_4_ sequentially changed from dimmed to clearly as the polarization state of the analyzer changed from RCP to LCP. Here, the phenomenon that the reconstructed image cannot completely disappear may mainly be caused by the fabrication error. The errors in structure parameters not only influence the phase but also bring a bigger difference between the R_LL_ and R_RR_ of each atom, which finally changes the combined reflection from quasilinear to elliptical polarized, and such a problem can be improved by optimizing the fabrication processes.

## 4. Metasurface for Spatially Varying Foci and Polarization States

Traditional lenses are based on the refraction scheme of geometric optics, the phase accumulation during the propagation of light. Thus, they are generally bulky and nonplanar, as the thickness of the lens is determined by the required phase distribution, which is usually curved. On the contrary, the phase profile of a metalens is controlled by the optical response of each meta-atom and, therefore, metalenses can obtain a curved phase profile with a planar and ultrathin configuration [4,5,6,32,33]. As fundamental optical elements, metalenses have shown promising applications in miniaturized and integrated optical systems for optical imaging, spectroscopic analysis and measurement, lithography, and so on.

In this paper, benefiting from the exotic properties of the proposed single-celled metasurfaces, a metalens array simultaneously providing focusing and multiple polarization state generation is proposed. As shown in Figure 4a, when an LP light incident on the metasurface, the reflection light will split into multiple beams, and the polarization state and focal point of each beam can be designed independently and arbitrarily. This feature can be used to generate light sources with multiple polarization states in the polarization measuring system. Based on such kinds of optical sources, results corresponding to multiple polarization states can be synchronously obtained through a single measurement, which can greatly improve the efficiency of testing. Moreover, it provides a solution for the problem of dynamic or living target measurement in the traditional time-sharing polarization system.

The desired phase profile for an ideal lens can be written as Equation (2) [4]:(2)$$\Phi \left(x,y\right)=\frac{2\pi }{\lambda }\left(\sqrt{(x^{2}+y^{2}+f^{2}})-f\right)$$
where *λ* is the operation wavelength, and *f* is the focal length. Metalens arrays and multifoci lenses are two common schemes to achieve multiple focal spots [4]. In this paper, the metalens array scheme was chosen and a four-metalens array with polarization rotation of 0°, 45°, 90°, and 120° was demonstrated to prove the validity of the strategy as shown in Figure 4b. Each metalens was composed of 50 × 50 single-celled meta-atoms. The operation wavelength and focal length were set as 1.55 μm and 35 μm, respectively. To combine the polarization rotation and focusing ability in the metalens, the corresponding phase profile can be expressed as Equations (3) and (4).
(3)$$\Phi_{RR}\left(x,y\right)=\frac{2\pi }{\lambda }\left(\sqrt{(x^{2}+y^{2}+f^{2}})-f\right)$$
(4)$$\Phi_{LL}\left(x,y\right)=\frac{2\pi }{\lambda }\left(\sqrt{(x^{2}+y^{2}+f^{2}})-f\right)+2\delta$$
where the values of *δ* were set as 0°, 45°, 90°, and 120° for the four lenses, respectively. In addition, to fully split the four beams as well as avoid the influence of the spin-preserving components, the focused beams of the four lenses were set off-axis with deflection angles of 10° by adding a corresponding phase gradient, as shown in Figure 3b.

The electric field distributions under the *x*-LP light of each metalens were simulated, and the far-field diffraction spots were calculated based on the vector diffraction theory. As shown in Figure 4c, the simulated focal spots showed obvious polarization features. By extracting the spot intensities in the *x* and *y* polarization directions, the polarization states of the four emergent lights could be calculated according to the *x*-pol and *y*-pol components of the spot. The results were quite close to 0°, 45°, 90°, and 120°, which also well match the design values. Figure 4d shows the intensity distribution of the focal spot along the *y*-axis with a 1.55 μm *x*-LP light incident on L_2_. The full-width at half-maximum (FWHM) was 1.57 μm, indicating the designed metalenses have pretty good focusing performance.

The intensity distribution of the designed lenses at different wavelengths was also simulated to show the broadband operation property. As shown in Figure 5, the lenses showed favorable focusing characteristics at 1.55 μm and 2 μm. The focus length of the four designed lenses under the incidence of 1.55 μm were, respectively, 31.5 μm, 33.1 μm, 33.6 μm, and 36.2 μm, which are quite close to the designed focal length *f* = 35 μm. This difference in focal length was mainly caused by the limited size of the designed lens and the phase difference between the actual phase and the ideal phase. By further increasing the lens size and further optimizing the actual phase distribution, both the actual focal length accuracy and focusing performance can be improved. According to the relationship between the phase gradient and deflection angle, for the same phase gradient, the longer the wavelength, the larger the deflection angle. It should be noted here that the metalenses are chromatic. In this design, the phase gradients for focusing and deflection were designed with a center wavelength of 1.55 μm; thus, compared with the results corresponding to 1.55 μm, the focal lengths under the incidence of 2 μm were much shorter (about 23 μm), as shown in Figure 5. The metalenses can also be designed achromatically by using some optimization algorithms [5]. Moreover, Tsilipakos et al. recently proposed an ingenious approach for the phase compensation of different frequencies by utilizing trains of multiple resonances [34]; such a strategy may also inspire an achromatic design of the proposed metalenses.

## 5. Conclusions

In this paper, a single-celled metasurface composed of chiral umbrella-shaped MIM unit cells is proposed for multiple polarization generation and wavefront shaping. The unit cells were designed as broadband and high-performance polarization-preserving mirrors, which provide the basic condition for wavefront shaping based on the geometric phase. By introducing the manipulation of the AA geometric phase, the independent manipulation of the phase and phase retardation of the spin-flipped components can be realized simultaneously and independently in a single unit cell. The above features finally contributed to independent control of the wavefront as well as the polarization state of reflection in a single-celled metasurface. To prove the validity of the proposed scheme, a vectorial hologram generator and metalens array with spatially varying polarizations were demonstrated, and both the simulated and experimental results agreed well with the design. Compared with the previous work based on the PB geometric phase, the proposed scheme in this paper can effectively avoid the pixel and efficiency losses caused by the intrinsic symmetry of the PB geometric phase. In addition, by virtue of the advantage of spin-independent control of the phase, no additional complicated algorithms were needed in the design. We believe this work will provide a new idea in the design of optical elements applied in miniaturized and integrated multipolarization-involved display, real-time imaging, or spectroscopy systems.

## Figures and Tables

**Figure 1 nanomaterials-12-04336-f001:**
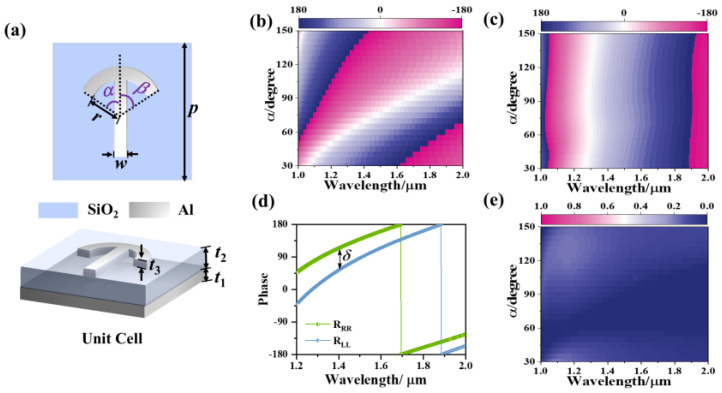
(**a**) Schematic diagram of the unit cell (**b**,**c**) phase of spin-flipped reflection. (**b**) φ_LL_ and (**c**) φ_RR_ varies with central angle *α* (**d**) phase retardation between φ_LL_ and φ_RR_ in the range of 1–2 µm when *α* = 40° and *β* = 60°. (**e**) Reflection difference between spin-flipped reflection R_LL_ and R_RR_ varies with central angle α. In these simulations, *p* = 700 nm, *r* = 120 nm, *w* = 80 nm, *t*_1_ = 200 nm, *t*_2_ = 180 nm, *t*_3_ = 100 nm. Specifically, in (**b**,**c**,**e**), *α* varies from 30° to 150° and *β* = 60°.

**Figure 2 nanomaterials-12-04336-f002:**
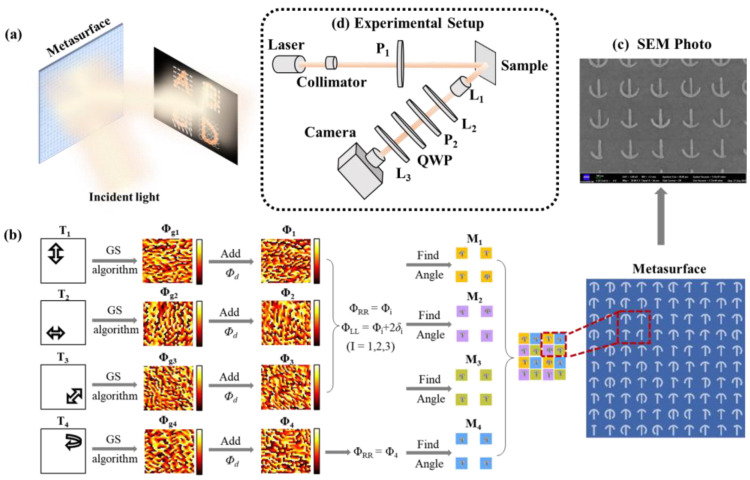
(**a**) Schematic diagram of vectorial holography with spatially varying polarization. (**b**) The design process of the metasurface for vectorial metaholography. (**c**) SEM photo of the fabricated sample. (**d**) Experimental setup for vectorial hologram polarization-sorting process (P_1_–polarizer 1, P_2_–polarizer 2, L_1_–objective lens, L_2_–lens 2, L_3_–lens 3, QWP–quarter waveplate. In the test, QWP is only used for the sorting of circular polarization).

**Figure 3 nanomaterials-12-04336-f003:**
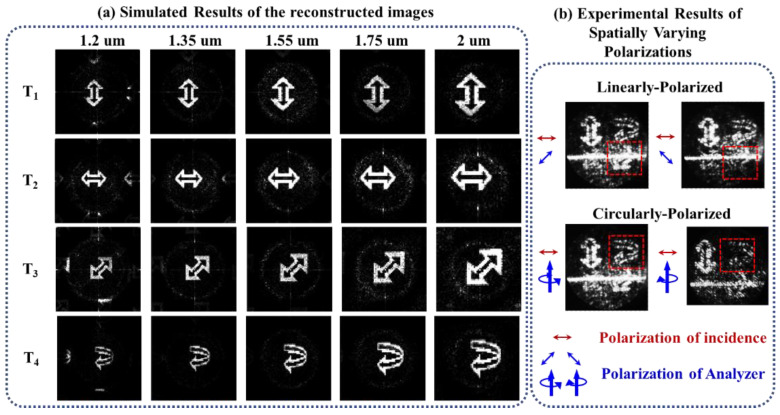
(**a**) Simulated results of reconstructed images at different wavelengths. (**b**) Experimental results of sorting spatially varying polarizations (*λ* = 1.55 µm).

**Figure 4 nanomaterials-12-04336-f004:**
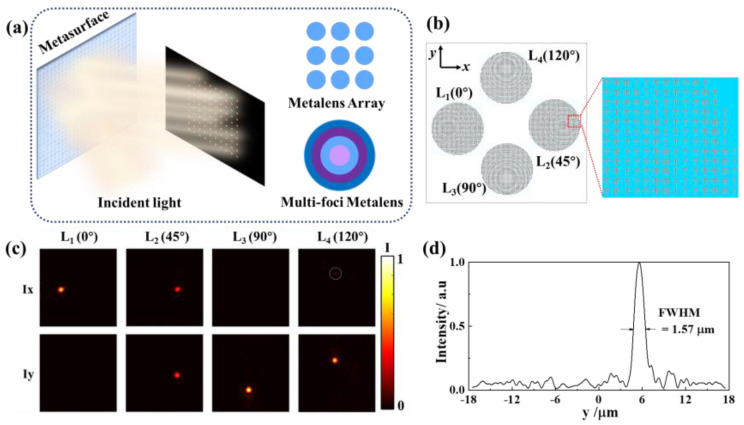
(**a**) Schematic diagram of metasurface with spatially varying focal spots and polarization states. The right two inserts show the two design schemes for obtaining spatially varying focal spots. (**b**) Schematic diagram of the designed metalens array. (**c**) Simulated *x* and *y* components of the focal spot intensity on the *x*–*y* plane (*z* = 33 μm); the generated polarization states are designed as 0°, 45°, 90°, and 120° for L_1_, L_2_, L_3_, and L_4_, respectively. (**d**) The intensity distribution of focal spot (L_2_, 1.55 μm) along the *y*-axis (*x* = 0 μm, *z* = 33 μm).

**Figure 5 nanomaterials-12-04336-f005:**
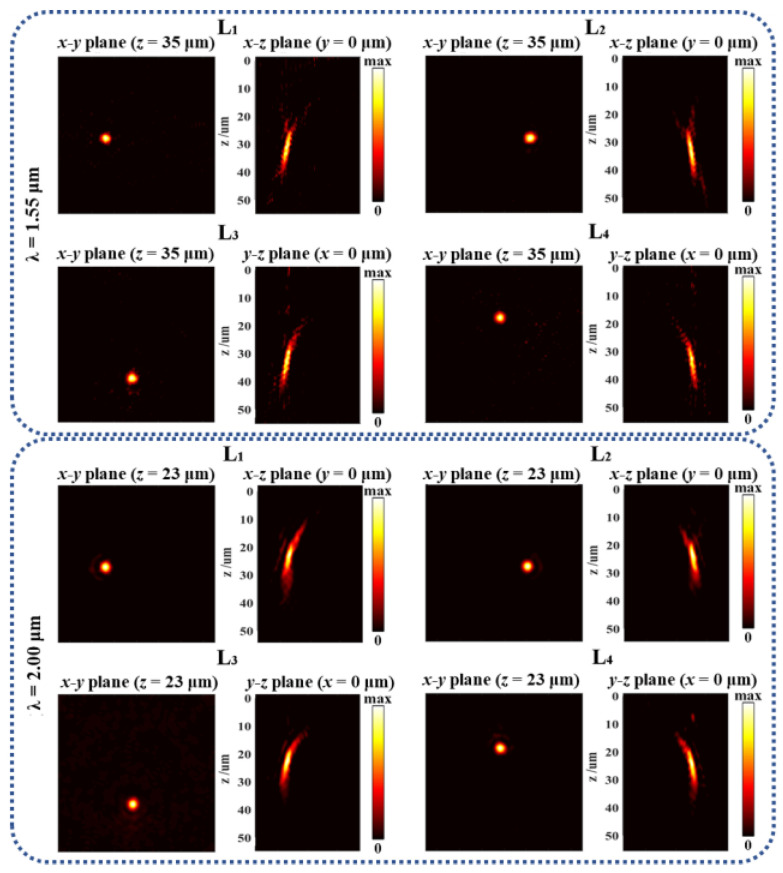
Intensity distributions of the designed lenses under incidence with different wavelengths.

## Data Availability

Not applicable.

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
