# Peer review of "A Single-Celled Metasurface for Multipolarization Generation and Wavefront Manipulation"

_nanomaterials, 2022, doi:10.3390/nano12234336_

Round 1

Reviewer 1 Report

Ji et al. have shown multi-polarization generation and wavefront shaping in the proposed umbrella like metasurface. The work is not at all novels. In fact the author published similar work in ACS Photonics 2021, 8, 1847−1855. I don’t see any significant novelty although just a different application has been projected. Hence the paper should be rejected without further consideration. I have some technical comments that might help the authors to improve the work and publish elsewhere.

1.     The fabrication explained by the author looks sketchy. The author mentioned they fabricated umbrella like metasurface via EBL which are on a dielectric of SiO2. I wonder how the author could do that in an EBL on a dielectric substrate since there would be charging effect which would make it almost impossible.

2.     The author should explain the abbreviation used in the manuscript like RLL and RRR . It makes confusing for the reader.

3.     What is the role of MIM over the bare metasurface on SiO2 or Si? Author should explain this in details and compare the results.

Author Response

The authors are grateful for the reviewer’s time and comments. We have scrutinized our manuscript and made some changes according to your valuable suggestions. The amendments are highlighted in the revised manuscript. Point by point responses to the reviewer’s comments are listed below.

Reviewer 2 Report

Line 16: change "lights" to "light"

Line 87: "the following equation" should read "Equation 1"

Line 191: "written as [4]" should read "written as Equation 2 [4]"

Line 192: This should be (2) and not (4)

Line 206: This should read Figure 4b

Line 209: This should read Figure 4c

Line 213: This should read Figure 4d

Line 200: "expressed as follows" should read "expressed as shown in Equations (3) and (4).

I think Materials and Methods should be an earlier section. It did not feel correct after the conclusions at the end of the paper.

Author Response

The authors are grateful for the reviewer’s time and comments. We have scrutinized our manuscript and made some changes according to your valuable suggestions. The amendments are highlighted in the revised manuscript. Please find the point-by-point responses in the attachment.

Reviewer 3 Report

In the manuscript nanomaterials-2057241 “A Single-Celled Metasurface for Multi-polarization Generation and Wavefront Manipulation” the authors study theoretically and experimentally metasurfaces based on the Aharonov-Anandam phase for polarization and wavefront control. The results may be useful to the community, since the study of flat metasurface-based optics for polarization and wavefront control is topical. The following comments should be addressed before publication.

1. The Aharonov-Anandam phase is a generalization of the PB phase; the latter is better known in metasurface literature. I suggest that the authors provide a short description of the difference between them to aid the reader.

2. In the abstract, acronyms MIM, PB, LCP and RCP are not defined.

3. From Fig. 5, it can be seen that the wavefront manipulation operation is achieved for 1550 and 2000 nm.  However, the focal length is quite different, i.e., the operation may be broadband but it is not achromatic, see e.g. DOI: 10.1002/adom.202000942. The authors are kindly invited to comment and bring this detail to the reader’s attention.

4. In the caption of Fig. 1, it is the phases of the reflection coefficients (\phi_LL and phi_RR) that are shown in panels (b) and (c), not R_LL and R_cc.

5. Caption of Fig. 4, lines 233-234. The description of panel (c) seems wrong.

6. In the Supporting Information, line 396, the reference is missing.

7. As a more general comment, in my opinion the level detail in describing the state of the art, the methodology, and the results could be improved, making it easier to follow for the reader.

Author Response

The authors are grateful for the reviewer’s time and comments. We have scrutinized our manuscript and made some changes according to your valuable suggestions. The amendments are highlighted in the revised manuscript. Please find the point-by-point response in the attachment.

Round 2

Reviewer 1 Report

Again, I still don't see any novelty of this paper. The author didn't work enough on the manuscript .  The technical details has lots of flaws. I don't think it should be publishable in a high impact factor journal. The author may submit it elsewhere.